# The structure of the proteins of Camp Hill virus

Sunil Thomas [ID] *

Lankenau Institute for Medical Research, Wynnewood, Pennsylvania, United States of America

* thomass-02@mlhs.org, suntom2@gmail.com

## Abstract

Camp Hill virus (CHV), a newly identified henipavirus, was recently discovered in northern short-tailed shrews in Camp Hill, Alabama. This marks a significant event as it is the first henipavirus ever reported in North America. The significance of henipaviruses lies in their ability to cause severe and often fatal diseases, such as encephalitis and respiratory illness, with a high mortality rate in both humans and animals. The emergence of new henipavirus strains, like CHV, amplifies concerns about the possibility of future zoonotic spillovers—where diseases are transmitted from animals to humans. Because henipaviruses can be highly contagious and have no specific antiviral treatment, their emergence poses a potential threat to public health. The major proteins of CHV include attachment glycoprotein, fusion protein, X protein, C protein, matrix protein, nucleocapsid protein, phosphoprotein, and RNA polymerase. In our study, we focused on determining the three-dimensional structure of these major proteins, providing crucial insights into how they function at the molecular level. Understanding the precise structure of these proteins is vital, as it can inform efforts to block the virus's ability to infect cells. Proteomic analysis confirmed that the proteins of CHV is similar to the proteins of Sollieres shrew parahenipa virus, Ninorex virus, Melian virus, Lechodon virus and Langya virus. We identified the B-cell and T-cell epitopes of these proteins. By characterizing these epitopes, our research contributes to the design of targeted vaccines that could stimulate a robust immune response against CHV. The identification of these epitopes also allows us to understand how the virus interacts with the immune system, which is essential for designing vaccines that can elicit both humoral and cellular immunity. Our study would lead to development of novel vaccines to protect against CHV.

## Introduction

Henipavirus infections are severe, potentially fatal illnesses that primarily affect the respiratory system and the brain, often leading to encephalitis [1]. These viruses belong to the *Paramyxoviridae* family and are negative-sense RNA viruses, with bats acting as their primary natural hosts. Henipaviruses are zoonotic, that can be

**Data availability statement:** All relevant data are within the paper. The model PDB files generated are not made in other public databases.

**Funding:** The author(s) received no specific funding for this work.

**Competing interests:** The author declares no conflict of interest.

transmitted from animals to humans and are capable of crossing species barriers to infect various mammals, including humans. Infected individuals may experience severe respiratory distress and neurological complications, with many cases proving fatal due to encephalitis or respiratory failure [1–3].

The two most prominent henipaviruses are Hendra virus and Nipah virus [4]. Hendra virus, first identified in Australia in 1994, has caused several outbreaks, with mortality rates reaching up to 70%. The virus is primarily transmitted from bats to horses, which can then infect humans, though direct human-to-human transmission has also been reported [5]. Nipah virus, first recognized during an outbreak in Malaysia in 1998–1999, has caused numerous outbreaks in Southeast Asia, particularly in Bangladesh and India, with case-fatality rates ranging from 40% to 75%, depending on local surveillance and healthcare quality [5]. Controlling animal reservoirs and limiting human exposure are essential components of public health efforts in regions affected by henipavirus infections.

Henipaviruses are highly significant due to their high fatality rates, leading to severe morbidity and mortality during documented outbreaks. Beyond the devastating human toll, these epidemics have substantial economic consequences. The destruction caused by outbreaks can overwhelm healthcare systems, disrupt local economies, and require costly containment measures [6].

The risk of henipavirus outbreaks is increasing due to human activities such as deforestation, which forces greater interaction between humans and bats, the primary zoonotic reservoir hosts. Climate change further alters habitats, potentially introducing the virus to new regions. Additionally, globalization and international trade contribute to the spread of disease, as demonstrated by Nipah virus outbreaks in Singapore and Malaysia [6].

In 2018, the Langya virus, a new henipavirus, was reported in humans in the Shandong and Henan provinces of eastern China. Phylogenetically related to Mojiang henipavirus, a rat-borne virus first identified in southern China in 2012, Langya virus is believed to have been transmitted from shrews to humans [7–9].

In 2021, researchers conducting a study on mammalian longevity made a significant discovery when they captured four northern short-tailed shrews (*Blarina brevicauda*) in the wild at Camp Hill, Auburn, Alabama, USA [3]. The researchers observed notable differences in the shrews' viral strains compared to those of closely related species, including both host species and geographical variations. Consequently, the researchers named the putative virus Camp Hill virus (CHV), reflecting its location of discovery. Interestingly, viral RNA from the captured shrews was only detected in their kidney tissues, strongly suggesting that CHV exhibits renal tropism.

This manuscript offers an in-depth analysis of the protein structure of CHV, highlighting the spatial organization of its individual protein components. The research goes beyond basic structural mapping by identifying and characterizing both B-cell and T-cell epitopes within the viral proteins. These epitopes are essential because they serve as key targets for the immune system, triggering both humoral (antibody-mediated) and cellular immune responses. Understanding the precise positioning and structure of these epitopes is invaluable for several reasons, particularly in the design of targeted diagnostic tools and vaccines.

By mapping the protein architecture and identifying specific immune system targets, the researchers can now work towards developing diagnostics that are more accurate and sensitive in detecting CHV infections. Furthermore, this detailed structural knowledge forms the foundation for the design of vaccines that can induce a strong and specific immune response, offering better protection against CHV and its associated diseases. The ability to pinpoint these critical epitopes enhances our capacity to prevent and manage CHV infections, making this research a pivotal step in advancing both therapeutic and preventive strategies against this novel virus.

## Materials and methods

### Protein sequence of Camp Hill virus

The CHV sequences were obtained from the NCBI protein database (https://www.ncbi.nlm.nih.gov/protein/) [10]. Table 1 lists the CHV proteins, their accession numbers, and associated functions.

### Protein modeling

A thorough understanding of biological systems requires insight into the functioning of protein complexes and networks, which necessitates a detailed analysis of protein interactions and their quaternary structure. Protein plots, also known as snake diagrams, provide a two-dimensional representation of a protein sequence, emphasizing features like secondary structure [11]. To generate these diagrams, we used Protter (http://wlab.ethz.ch/protter) [12], an interactive web-based tool. Protter enables users to visualize and integrate annotated and predicted protein sequence features, experimental proteomic data, and post-translational modifications within the protein's transmembrane topology. It offers options to select from various annotation sources, upload proteomics data files, choose peptides for targeted quantitative proteomics, and create high-quality visualizations [13].

Phobius was utilized for the prediction of transmembrane topology and signal peptides from the amino acid sequence of proteins (https://phobius.sbc.su.se/) [14,15]. This tool is designed to analyze the structure of proteins and predict whether specific regions of a protein are embedded within the membrane or function as signal peptides, which are crucial for understanding protein localization and function.

For three-dimensional homology modeling, we employed AlphaFold version 2 [16]. The tool was used with default settings, and the resulting models were visualized using RasMol [16]. The protein sequences of CHV were submitted in FASTA format for these analyses. AlphaFold, a cutting-edge deep learning tool, is known for producing highly accurate structural predictions based on sequence data, whereas Phyre2 provides reliable homology-based modeling using iterative threading techniques.

In addition to AlphaFold version 2, we also used the Iterative Threading Assembly Refinement (I-TASSER) (https://zhanglab.ccmb.med.umich.edu/I-TASSER/) [17] for three-dimensional homology modeling, again using default settings.

**Table 1. The Camp Hill virus (CHV) major proteins, accession numbers and its function.**

| CHV Protein | Accession Number | Function |
| --- | --- | --- |
| Attachment glycoprotein | XJU75835.1 | Binds host cell receptor |
| Fusion protein | XJU75834.1 | Membrane fusion |
| X protein | XJU75833.1 | Unknown |
| C protein | XJU75831.1 | Viral budding, IFN antagonist |
| Matrix protein | XJU75832.1 | Viral assembly, budding, interferon antagonist |
| Nucleocapsid protein | XJU75830.1 | Encapsidation, replication |
| Phosphoprotein | XJU75829.1 | RNA transcription, replicatioon |
| RNA polymerase | XJU75836.1 | RNA synthesis, replication |

The protein sequences of CHV were entered in FASTA format. I-TASSER is a widely used tool that applies threading to align proteins against known templates, refining the models iteratively to improve their accuracy. For each protein model generated, we determined several metrics, including the confidence score (C-score), template modeling score (TM-score), Root Mean Square Deviation (RMSD), and normalized B-factor, which provided insights into the model's accuracy and reliability.

The normalized B-factor, or B-factor profile (BFP), is derived by applying a z-score normalization to the raw B-factor values. The B-factor itself represents the flexibility or mobility of atoms or residues within a protein, indicating how much thermal motion occurs at each position. A higher B-factor typically indicates greater flexibility, while lower values suggest more rigidity. The BFP, which is predicted through a combination of template-based assignments and profile-based prediction methods, helps in identifying regions of the protein that may be less stable in experimental structures. Residues with a BFP greater than 0 are generally considered less stable [18].

The C-score, a crucial confidence metric for I-TASSER, assesses the overall quality of a predicted model. It is calculated based on the significance of the threading template alignments and the convergence of the structure assembly simulations. The C-score typically ranges from −5 to +2, with higher values indicating greater confidence in the model's accuracy [19]. This score is essential in evaluating the trustworthiness of the predictions, especially when the experimental structure is unavailable.

TM-score and RMSD are widely recognized metrics for evaluating the structural similarity between predicted models and native protein structures. The TM-score measures the global structural similarity, and RMSD calculates the average distance between corresponding atoms in two structures. These metrics are particularly useful for assessing model accuracy when the native structure is known, although when the native structure is unknown, they help predict model quality based on other structural features [20,21].

To predict the B-cell and T-cell epitopes (specifically the MHC-II allele HLA-DRB1), we used the IEDB Analysis Resource (http://tools.iedb.org) [22], which offers powerful tools for predicting potential immune epitopes within the protein sequences.

Finally, the predicted solvent accessibility of the protein was determined using NetSurfP-3.0 (https://services.health-tech.dtu.dk/services/NetSurfP-3.0/) [23]. NetSurfP-3.0 is an advanced tool designed to predict several important features of proteins, including solvent accessibility, secondary structure, structural disorder, and backbone dihedral angles for individual residues. The latest version integrates pre-trained protein language models, which significantly enhance the performance and speed of the predictions—achieving a dramatic reduction in runtime and improving the tool's performance by two orders of magnitude over previous versions, while maintaining high prediction accuracy [24].

Percentage identity of the proteins of CHV were compared using protein BLAST (https://blast.ncbi.nlm.nih.gov/Blast.cgi?PROGRAM=blastp&PAGE_TYPE=BlastSearch&LINK_LOC=blasthome) [25].

## Results

The core proteins of CHV encompass several key functional components, including the attachment glycoprotein, fusion protein, X protein, C protein, matrix protein, nucleocapsid protein, phosphoprotein, and RNA polymerase. These proteins play crucial roles in the viral life cycle, contributing to processes such as viral attachment, entry, replication, and assembly.

Our study revealed that certain proteins, specifically the attachment glycoprotein, fusion protein, X protein, and matrix protein, possess transmembrane characteristics, making them integral membrane proteins. These transmembrane proteins are of particular interest in the context of vaccine development due to their potential to trigger immune responses. The attachment glycoprotein and fusion protein are key players in viral entry, facilitating the virus's interaction with host cell receptors and the fusion of viral and host cell membranes. The X protein and matrix protein, on the other hand, are involved in various aspects of viral assembly and stability, contributing to the viral structural integrity and its ability to infect host cells.

Transmembrane proteins like these are attractive targets for vaccine development, as they are often exposed on the surface of the virus and are critical for the virus's ability to infect host cells. Targeting these proteins with vaccines could potentially block viral entry and prevent infection, making them promising candidates for further research and therapeutic intervention. Additionally, the immunogenic potential of these proteins suggests they could stimulate both humoral and cellular immune responses, which are essential for an effective immune defense against viral infections.

### Attachment glycoprotein

The entry of henipaviruses into host cells is critically dependent on the attachment glycoprotein, which initiates infection by binding to host cell receptors [26]. In the case of Camp Hill virus (CHV), a novel member of the *Henipavirus* genus, the attachment glycoprotein—located on the outer surface of the virion—serves as the major structural protein responsible for facilitating viral entry into host cells.

To characterize this key viral component, bioinformatic tools were employed to analyze its structural features. Initial analysis using Protter identified a single transmembrane domain within the attachment glycoprotein, suggesting its anchoring within the viral envelope. This structural topology was further validated using Phobius, which confirmed the presence and position of the transmembrane region (Fig 1A, B).

To investigate the three-dimensional architecture of the protein, AlphaFold 2 was used to generate a predicted model of the CHV attachment glycoprotein (Fig 1C). The predicted structure provides insights into potential functional domains and spatial organization relevant to receptor interaction and immune recognition.

Immunoinformatic predictions identified multiple B-cell and T-cell epitopes within the protein sequence, as shown in Fig 1D. Notably, several of these epitopes overlap, suggesting regions of high immunogenic potential that may serve as promising targets for vaccine development or immunotherapeutic interventions.

Further structural characterization was performed by analyzing the normalized B-factor distribution (Fig 1E), which provides information on atomic flexibility across the protein. Additionally, predicted solvent accessibility data (Fig 1F) highlighted surface-exposed regions that are likely to be accessible to antibodies and immune effectors.

The quality and reliability of the predicted protein model were assessed using structural validation metrics, including the confidence score (C-score), template modeling (TM-score), and root mean square deviation (RMSD), which are summarized in Table 2. RMSD is a metric used to quantify the average distance between the atomic positions of protein structures, typically to assess structural changes over time during molecular simulations. Higher RMSD values indicate greater structural deviation, while an RMSD of zero denotes identical conformations [27]. These comprehensive analyses contribute to a deeper understanding of the CHV attachment glycoprotein's structure and immunogenic properties, supporting its potential as a candidate for therapeutic or vaccine design.

### Fusion protein

Henipaviruses entry and fusion requires the coordinated action of both the fusion protein and attachment glycoprotein [28]. Analysis using Protter identified two transmembrane domains within the fusion protein, suggesting its anchoring within the viral envelope. The structural topology was further validated using Phobius, which confirmed the presence and position of the transmembrane region (Fig 2A, B).

To investigate the three-dimensional architecture of the protein, AlphaFold 2 was used to generate a predicted model of the CHV fusion protein (Fig 2C). The predicted structure reveals potential functional domains and spatial organization that may play key roles in receptor interaction and immune recognition.

Immunoinformatic predictions identified multiple B-cell and T-cell epitopes within the protein sequence, as shown in Fig 2D. Further structural characterization was performed by analyzing the normalized B-factor distribution (Fig 2E). The

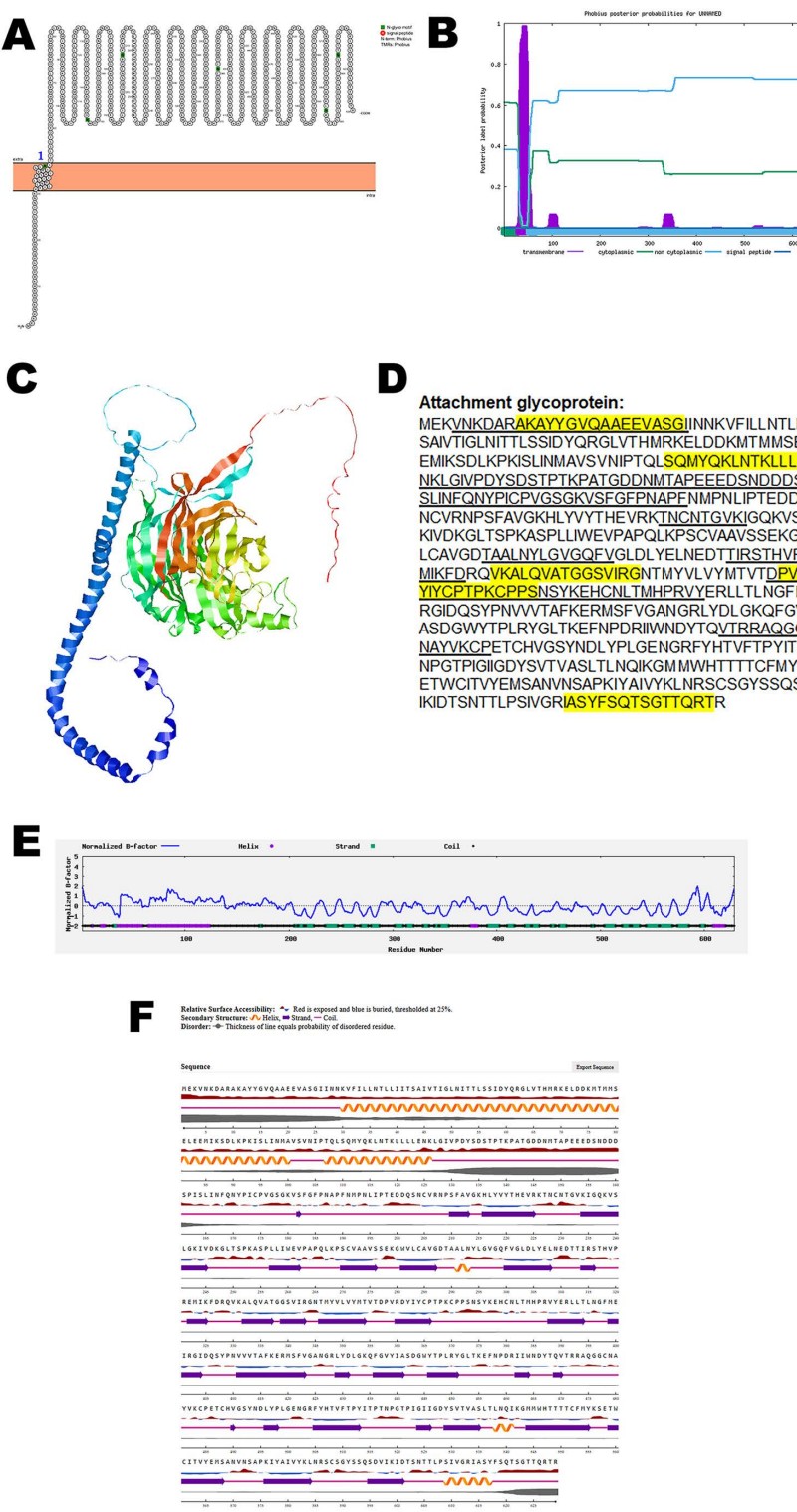

**Fig 1. The predicted structure of attachment glycoprotein of CHV.** (A). The topology of CHV attachment glycoprotein as determined by Protter. (B). The topology of CHV attachment glycoprotein as determined by Phobius. (C). The predicted protein structure of CHV attachment glycoprotein (ribbon diagram) determined using the software AlfaFold 2. (D). The B-cell (underlined) and T-cell epitope (yellow shaded) of CHV attachment glycoprotein (Accession XJU75835.1). (E). Normalized B-factor of CHV attachment glycoprotein determined using I-TASSER. (F). The predicted solvent accessibility of attachment glycoprotein determined using NetSurfP-3.0.

**Table 2. The confidence score (C-score), template modeling (TM-score) and root mean square deviation (RMSD) of CHV proteins.**

| Protein | C-score | TM-score | RMSD |
|---|---|---|---|
| Attachment glycoprotein | −2.87 | 0.39±0.13 | 15.0±3.5Å |
| Fusion protein | −1.20 | 0.56±0.15 | 10.5±4.6Å |
| X protein | −2.08 | 0.47±0.15 | 9.3±4.6Å |
| C protein | −2.70 | 0.40±0.14 | 11.4±4.5Å |
| Matrix protein | 0.85 | 0.83±0.08 | 4.7±3.1Å |
| Nucleocapsid protein | −1.00 | 0.59±0.14 | 9.8±4.6Å |
| Phosphoprotein | −2.70 | 0.40±0.14 | 14.2±3.8Å |
| RNA polymerase | −0.27 | 0.68±0.12 | 10.5±4.6Å |

predicted solvent accessibility data (Fig 2F) highlighted surface-exposed regions that are accessible to antibodies and immune effectors. The C-score, TM-score, and RMSD of the predicted protein structure is summarized in Table 2.

## X protein

The role of X protein in the CHV is not clearly understood. Analysis using Protter identified one transmembrane domain, suggesting its anchoring within the viral envelope. The structural topology was further validated using Phobius, which confirmed the presence and position of the transmembrane region (Fig 3A, B).

To investigate the three-dimensional architecture of the protein, AlphaFold 2 was used to generate a predicted model of the CHV X protein (Fig 3C). The predicted structure provides insights into potential functional domains and spatial organization relevant to receptor interaction and immune recognition.

Immunoinformatic predictions identified multiple B-cell and T-cell epitopes within the protein sequence, as shown in Fig 3D. Further structural characterization was performed by analyzing the normalized B-factor distribution (Fig 3E). The predicted solvent accessibility data (Fig 3F) highlighted surface-exposed regions that are accessible to antibodies and immune effectors. The C-score, TM-score, and RMSD of the predicted protein structure is summarized in Table 2.

## C protein

The C protein of paramyxoviruses, including those of the Henipavirus genus, is a multifunctional accessory protein thought to contribute to various stages of the viral life cycle. One proposed role of the C protein is in the budding process, where it may assist in the assembly or release of matrix proteins, thereby facilitating efficient virion formation and egress from the host cell. In addition to its involvement in viral morphogenesis, the C protein is believed to participate in intracellular trafficking, potentially shuttling between the cytoplasm and the nucleus of infected host cells. This nucleocytoplasmic transport suggests a regulatory function, possibly linked to modulation of host cell processes. Furthermore, the C protein has been implicated in the suppression of host immune responses, particularly through interference with the interferon (IFN) signaling pathway. By antagonizing the host's innate immune defenses, the C protein may enhance viral replication and contribute to immune evasion, underscoring its importance in viral pathogenicity [29–31].

An in-depth structural analysis of the C protein using the Protter tool revealed the absence of any transmembrane domains, indicating that the protein is likely not embedded within virus membranes. To corroborate these findings, the predicted topology was independently validated using the Phobius server, which similarly confirmed the lack of transmembrane regions (Fig 4A, B). Together, these results support the conclusion that the C protein adopts a non-membranous conformation.

To investigate the three-dimensional architecture of the protein, AlphaFold 2 was used to generate a predicted model of the CHV C protein (Fig 4C). Immunoinformatic predictions identified multiple B-cell and T-cell epitopes within the protein sequence, as shown in Fig 4D. Further structural characterization was performed by analyzing the normalized B-factor

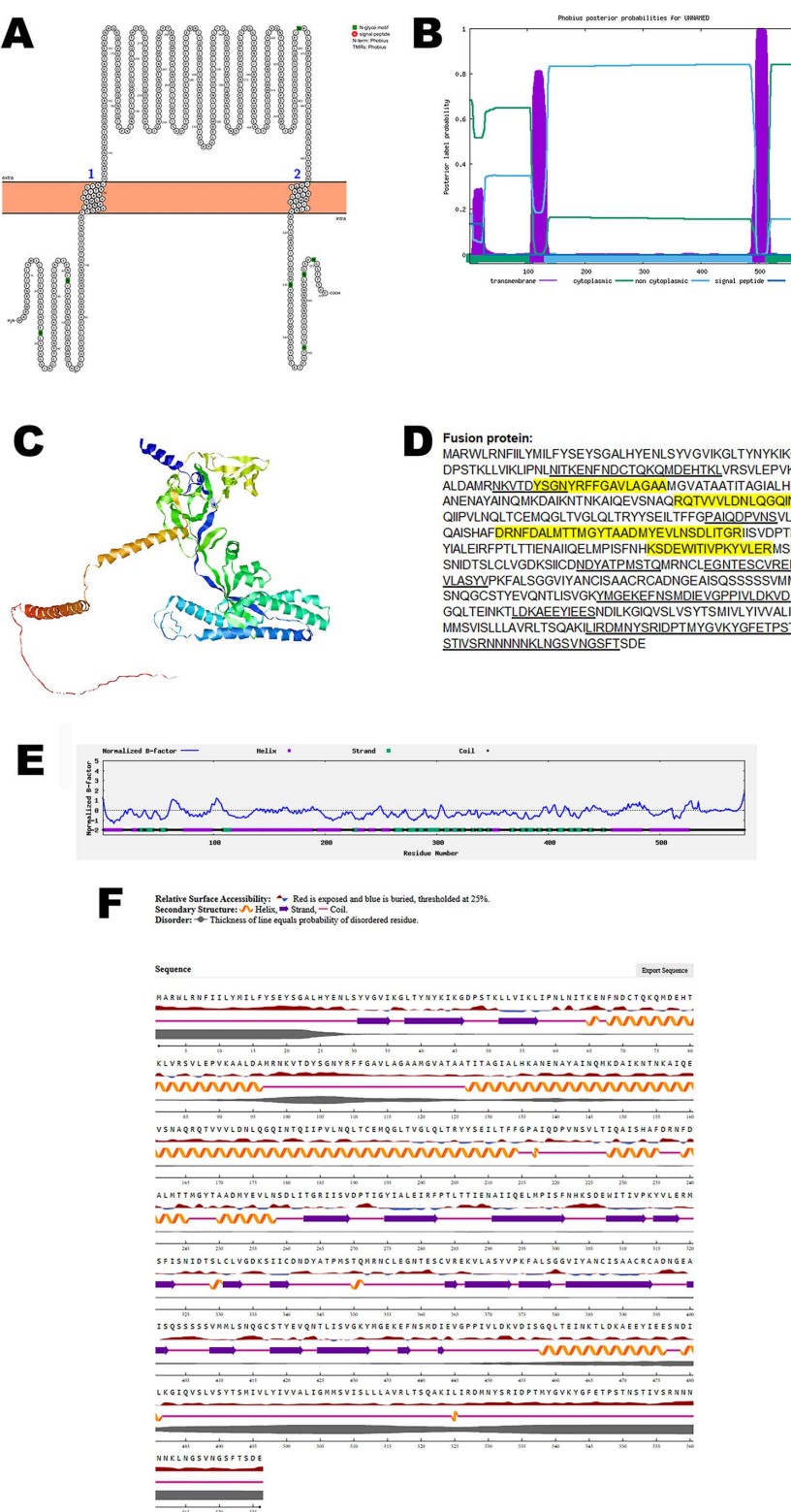

**Fig 2. The predicted structure of fusion protein of CHV.** (A). The topology of CHV fusion protein as determined by Protter. (B). The topology of CHV fusion protein as determined by Phobius. (C). The predicted protein structure of CHV fusion protein (ribbon diagram) determined using the software Alfa-Fold 2. (D). The B-cell (underlined) and T-cell epitope (yellow shaded) of CHV fusion protein (Accession XJU75834.1). (E). Normalized B-factor of CHV fusion protein determined using I-TASSER. (F). The predicted solvent accessibility of fusion protein determined using NetSurfP-3.0.

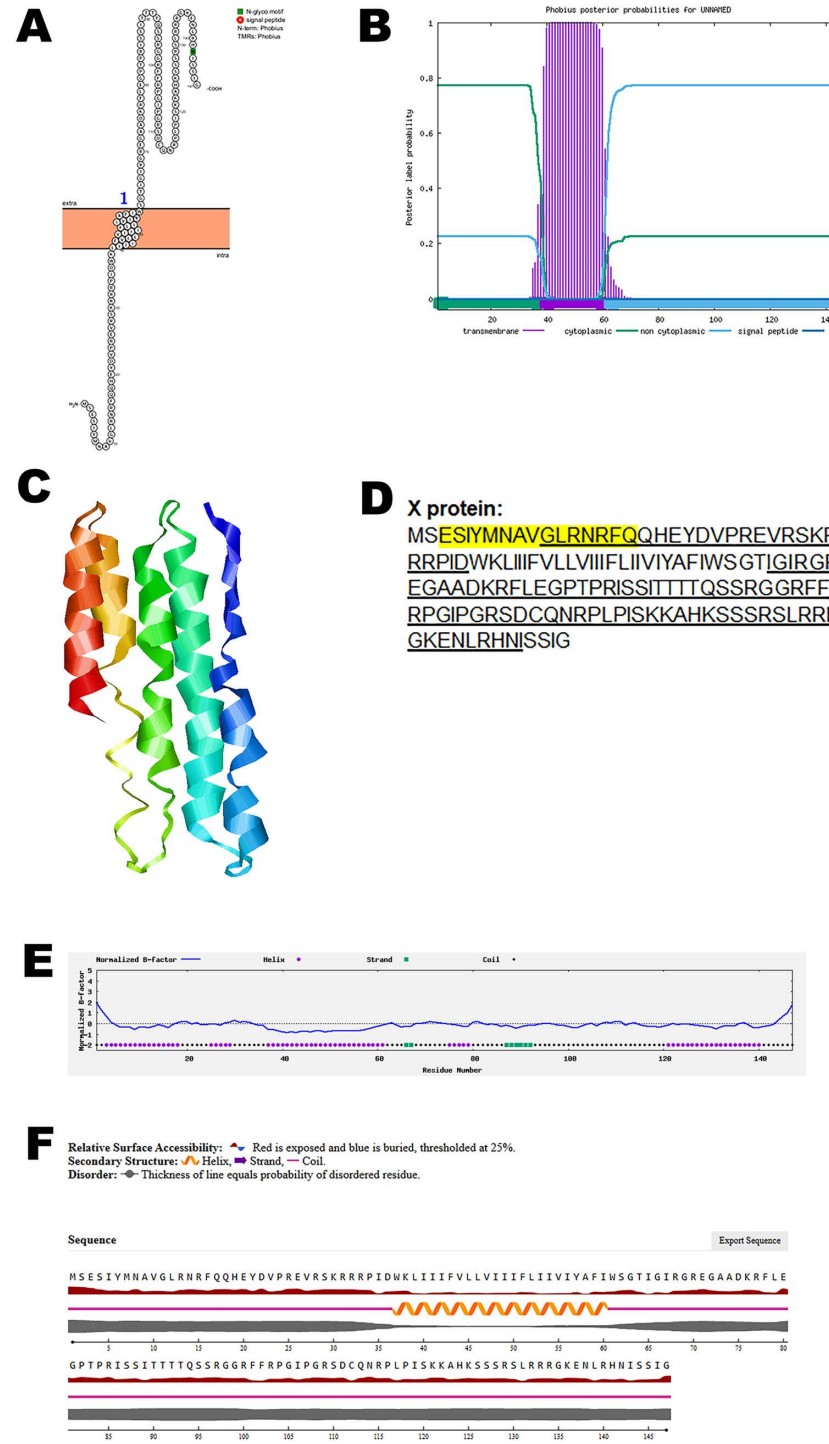

**Fig 3. The predicted structure of X protein of CHV.** (A). The topology of CHV X protein as determined by Protter. (B). The topology of CHV X protein as determined by Phobius. (C). The predicted protein structure of CHV X protein (ribbon diagram) determined using the software AlfaFold 2. (D). The B-cell (underlined) and T-cell epitope (yellow shaded) of CHV X protein (Accession XJU75833.1). (E). Normalized B-factor of CHV X protein determined using I-TASSER. (F). The predicted solvent accessibility of X protein determined using NetSurfP-3.0.

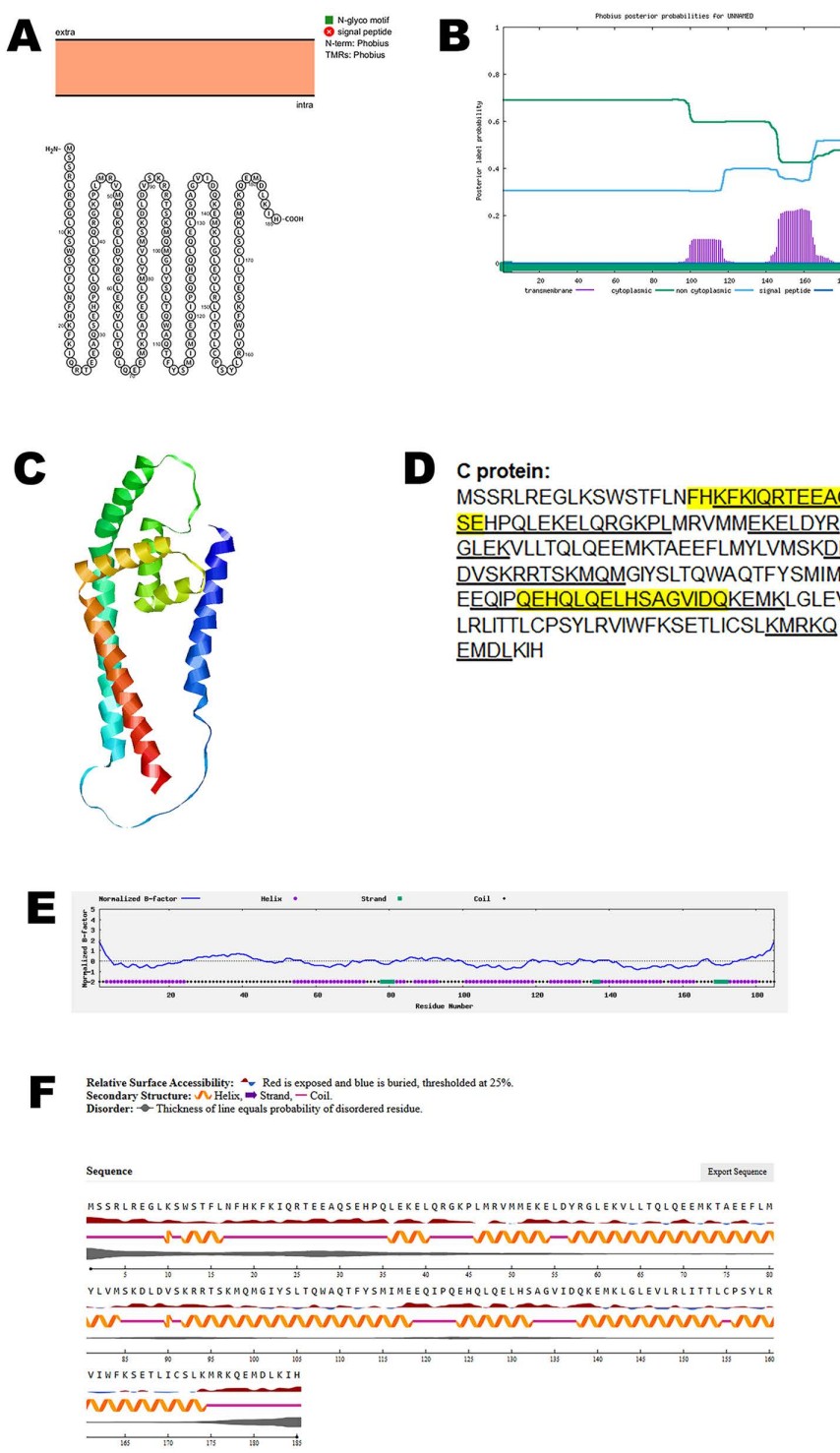

**Fig 4. The predicted structure of C protein of CHV.** (A). The topology of CHV C protein as determined by Protter. (B). The topology of CHV C protein as determined by Phobius. (C). The predicted protein structure of CHV C protein (ribbon diagram) determined using the software AlfaFold 2. (D). The B-cell (underlined) and T-cell epitope (yellow shaded) of CHV C protein (Accession XJU75831.1). (E). Normalized B-factor of CHV C protein determined using I-TASSER. (F). The predicted solvent accessibility of C protein determined using NetSurfP-3.0.

distribution (Fig 4E). The predicted solvent accessibility data (Fig 4F) highlighted surface-exposed regions that are accessible to antibodies and immune effectors. The C-score, TM-score, and RMSD of the protein structure is summarized in Table 2.

### Matrix protein

The matrix (M) protein of henipaviruses plays a central role in the assembly and budding of new viral particles, serving as a critical structural component of the virion. Beyond its structural functions, the M protein also exhibits non-structural roles, acting as an antagonist of the type I interferon response—a key element of the host's innate immune defense. Intriguingly, the M protein undergoes nuclear-cytoplasmic trafficking, a dynamic process that appears essential for its multifunctional nature. During its transit through the nucleus, the M protein is subject to monoubiquitination, a post-translational modification that is crucial for regulating its subsequent interactions. This modification facilitates proper cell sorting, membrane targeting, and the orchestration of budding at the plasma membrane, underscoring the complexity of M's involvement in both viral replication and immune evasion [32].

Analysis using Protter identified one transmembrane domain, suggesting its anchoring within the viral envelope. The structural topology was further validated using Phobius, which confirmed the presence and position of the transmembrane region (Fig 5A, B).

To investigate the three-dimensional architecture of the protein, AlphaFold 2 was used to generate a predicted model of the CHV matrix protein (Fig 5C). Immunoinformatic predictions identified multiple B-cell and T-cell epitopes within the protein sequence, as shown in Fig 5D. Further structural characterization was performed by analyzing the normalized B-factor distribution (Fig 5E). The predicted solvent accessibility data (Fig 5F) highlighted surface-exposed regions that are accessible to antibodies and immune effectors.

The quality and reliability of the predicted protein model were assessed using structural validation metrics, including the C-score, TM-score, and RMSD, which are summarized in Table 2.

### Nucleocapsid protein

The inner structural component of the henipavirus includes the nucleocapsid protein, the phosphoprotein, the RNA-dependent RNA polymerase, and viral genomic RNA (RNP complex). The RNP complex is essential and sufficient for the synthesis of viral RNA [33]. The nucleocapsid protein protects the RNA from degradation and facilitates its encapsidation. This structure is essential for both transcription and replication of the viral genome [33].

An in-depth structural analysis of the nucleocapsid protein using the Protter tool revealed the absence of any transmembrane domains, indicating that the protein is likely not embedded within virus membranes. To corroborate these findings, the predicted topology was independently validated using the Phobius server, which similarly confirmed the lack of transmembrane regions (Fig 6A, B). Together, these results support the conclusion that the nucleocapsid protein adopts a non-membranous conformation.

To investigate the three-dimensional architecture of the protein, AlphaFold 2 was used to generate a predicted model of the CHV nucleocapsid protein (Fig 6C). Immunoinformatic predictions identified multiple B-cell and T-cell epitopes within the protein sequence, as shown in Fig 6D. Further structural characterization was performed by analyzing the normalized B-factor distribution (Fig 6E). The predicted solvent accessibility data (Fig 6F) highlighted surface-exposed regions that are accessible to antibodies and immune effectors. The reliability of the predicted protein structure based on the C-score, TM-score, and RMSD, are summarized in Table 2.

### Phosphoprotein

Henipavirus uses its phosphoprotein to edit mRNA during transcription to evade the host's immune system. The P protein also plays a crucial role in RNA replication [34].

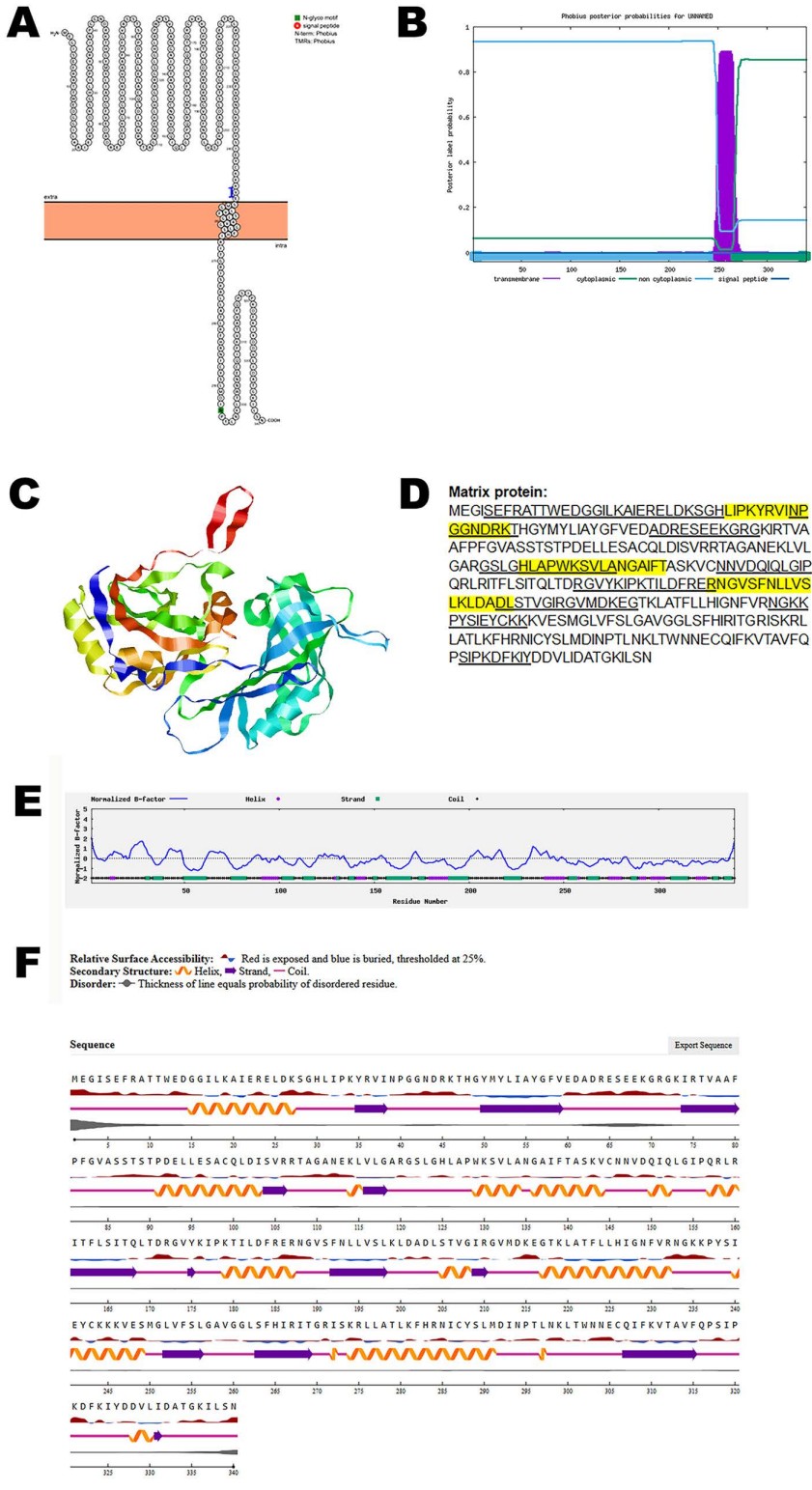

**Fig 5. The predicted structure of matrix protein of CHV.** (A). The topology of CHV matrix protein as determined by Protter. (B). The topology of CHV matrix protein as determined by Phobius. (C). The predicted protein structure of CHV matrix protein (ribbon diagram) determined using the software Alfa-Fold 2. (D). The B-cell (underlined) and T-cell epitope (yellow shaded) of CHV matrix protein (Accession XJU75832.1). (E). Normalized B-factor of CHV matrix protein determined using I-TASSER. (F). The predicted solvent accessibility of matrix protein determined using NetSurfP-3.0.

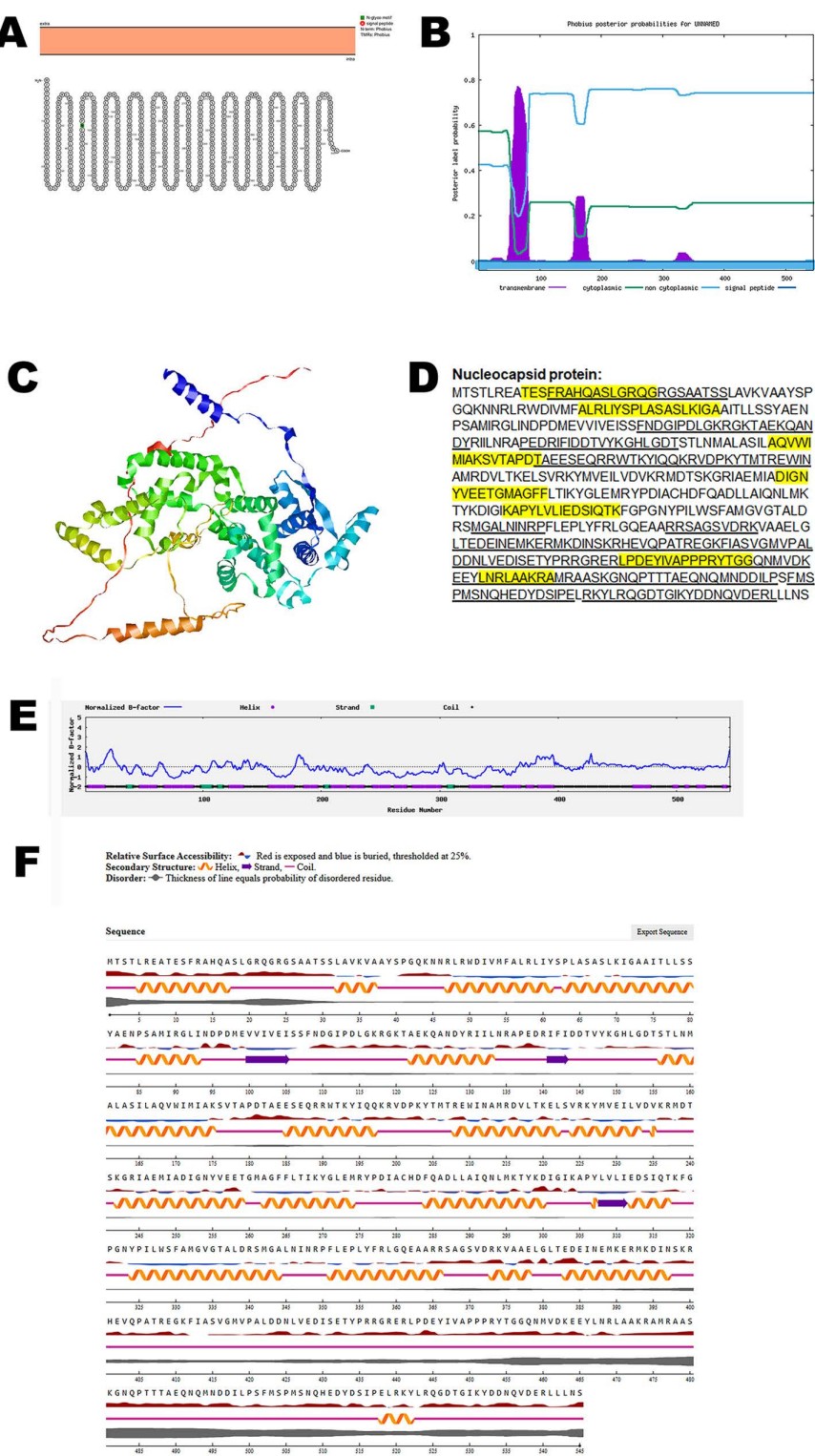

**Fig 6. The predicted structure of nucleocapsid protein of CHV.** (A). The topology of CHV nucleocapsid protein as determined by Protter. (B). The topology of CHV nucleocapsid protein as determined by Phobius. (C). The predicted protein structure of CHV nucleocapsid protein (ribbon diagram) determined using the software AlfaFold 2. (D). The B-cell (underlined) and T-cell epitope (yellow shaded) of CHV nucleocapsid protein (Accession XJU75830.1). (E). Normalized B-factor of CHV nucleocapsid protein determined using I-TASSER. (F). The predicted solvent accessibility of nucleocapsid protein determined using NetSurfP-3.0.

Structural analysis of the phosphoprotein using the Protter tool revealed the absence of any transmembrane domains, indicating that the protein is likely not embedded within virus membranes. To corroborate these findings, the predicted topology was independently validated using the Phobius server, which similarly confirmed the lack of transmembrane regions (Fig 7A, B). Together, these results support the conclusion that the phosphoprotein adopts a non-membranous conformation.

To investigate the three-dimensional architecture of the protein, AlphaFold 2 was used to generate a predicted model of the CHV phosphoprotein (Fig 7C). Immunoinformatic predictions identified multiple B-cell and T-cell epitopes within the protein sequence, as shown in Fig 7D. Further structural characterization was performed by analyzing the normalized B-factor distribution (Fig 7E). The predicted solvent accessibility data (Fig 7F) highlighted surface-exposed regions that are accessible to antibodies and immune effectors. Protein model quality was evaluated using C-score, TM-score, and RMSD, as summarized in Table 2.

### RNA polymerase

RNA polymerase is involved in replication of the CHV RNA. Structural analysis of the RNA polymerase using the Protter tool revealed the absence of any transmembrane domains. To corroborate these findings, the predicted topology was independently validated using the Phobius server, which similarly confirmed the lack of transmembrane regions (Fig 8A, B). Together, these results support the conclusion that the RNA polymerase adopts a non-membranous conformation.

To investigate the three-dimensional architecture of the protein, AlphaFold 2 was used to generate a predicted model of the CHV RNA polymerase (Fig 8C). Immunoinformatic predictions identified multiple B-cell and T-cell epitopes within the protein sequence, as shown in Fig 8D. Further structural characterization was performed by analyzing the normalized B-factor distribution (Fig 8E). The predicted solvent accessibility data (Fig 8F) highlighted surface-exposed regions that are accessible to antibodies and immune effectors.

The quality and reliability of the predicted protein models were evaluated using structural validation metrics such as the C-score, TM-score, and RMSD, with the results summarized in Table 2.

### Discussion

Henipaviruses are enveloped, single-stranded RNA viruses within the Paramyxoviridae family that can infect both humans and various animal species. Among them, Hendra virus (HeV) and Nipah virus (NiV) are of greatest concern due to their high virulence and frequent association with severe respiratory and neurological illnesses that carry high case-fatality rates. In August 2022, a novel species called Langya virus (LayV) was identified in eastern China in patients with fever and flu-like symptoms. Phylogenetic analysis shows LayV is closely related to Mojiang virus, another Henipavirus species previously found in rodents. While LayV can infect humans, its full pathogenic profile and transmission dynamics are still being studied. In contrast, Mojiang virus, Cedar virus (from Australia), and Ghanaian bat virus have not been linked to human disease so far, but ongoing research and surveillance remain important due to the risk of zoonotic spillover [35].

The identification of CHV, a newly recognized henipavirus in North America, is particularly significant due to the severe disease potential historically associated with other members of the *Henipavirus* genus, such as Nipah and Hendra viruses [3]. This is the first henipavirus reported in North America. These viruses are characterized by high case-fatality rates and the ability to cause outbreaks with serious public health implications. The CHV was observed in northern short-tailed shrews (*Blarina brevicauda*) highlighting the emerging role of this species as a potential zoonotic reservoir. These small mammals are now recognized as hosts for multiple viruses of public health concern, underlining the ecological complexity and potential risk they represent. CHV was detected in tissue samples from all northern short-tailed shrews analyzed, suggesting widespread circulation within this population [3].

Geographically, the northern short-tailed shrew is widely distributed across central and eastern North America, and its range often overlaps with human-populated areas. Their adaptability and tendency to enter human-modified environments

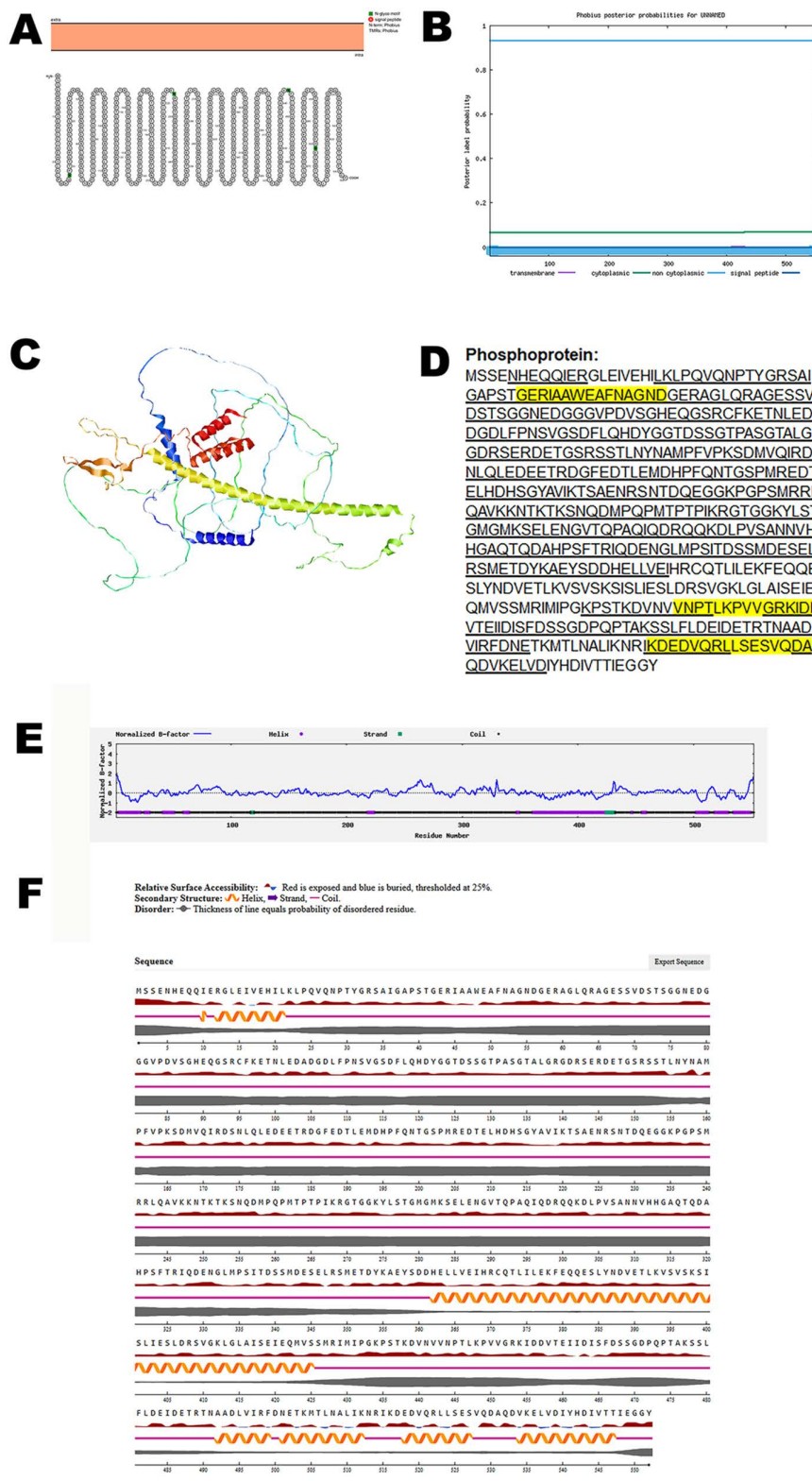

**Fig 7. The predicted structure of phosphoprotein of CHV.** (A). The topology of CHV phosphoprotein as determined by Protter. (B). The topology of CHV phosphoprotein as determined by Phobius. (C). The predicted protein structure of CHV phosphoprotein (ribbon diagram) determined using the software AlfaFold 2. (D). The B-cell (underlined) and T-cell epitope (yellow shaded) of CHV phosphoprotein (Accession XJU75829.1). (E). Normalized B-factor of CHV phosphoprotein determined using I-TASSER. (F). The predicted solvent accessibility of phosphoprotein determined using NetSurfP-3.0.

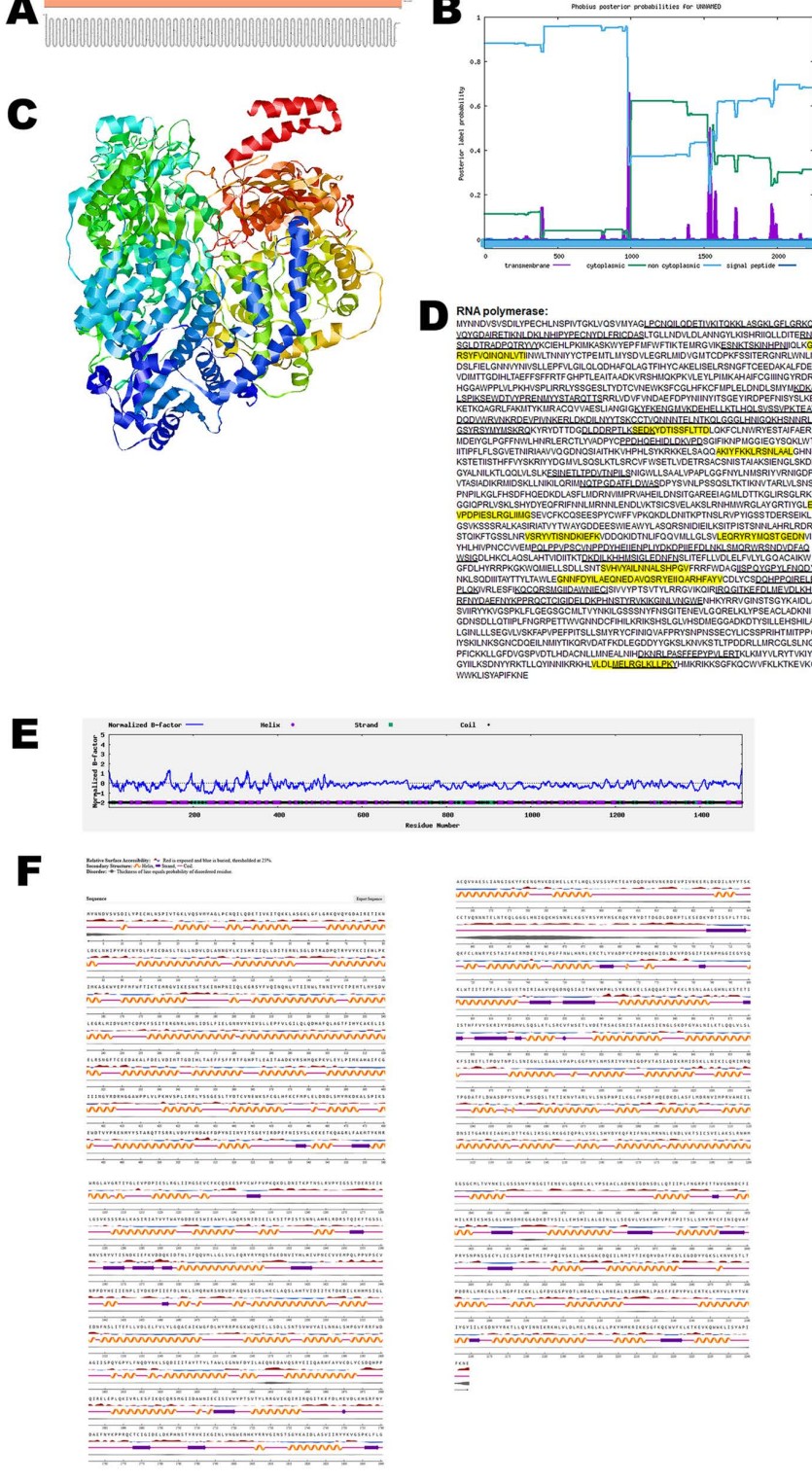

**Fig 8. The predicted structure of RNA polymerase of CHV.** (A). The topology of CHV RNA polymerase as determined by Protter. (B). The topology of CHV RNA polymerase as determined by Phobius. (C). The predicted protein structure of CHV RNA polymerase (ribbon diagram) determined using the software AlfaFold 2. (D). The B-cell (underlined) and T-cell epitope (yellow shaded) of CHV RNA polymerase (Accession XJU75836.1). (E). Normalized B-factor of CHV RNA polymerase determined using I-TASSER. (F). The predicted solvent accessibility of RNA polymerase determined using NetSurfP-3.0.

increase the likelihood of zoonotic spillover events, especially when contact with contaminated excreta or infected animals occurs.

Although CHV is not known to infect humans, direct contact or exposure to infected secretions or feces could present a risk. This underscores the urgent need for expanded surveillance, ecological studies, and risk assessments to evaluate the likelihood of human infection and understand the dynamics of CHV and other shrew-borne viruses. Proactively identifying transmission routes and implementing prevention strategies will be critical for reducing the risk of future outbreaks and enhancing preparedness for emerging zoonotic threats.

Northern short-tailed shrew harbors other viruses including the parahenipavirus, Blarina brevicauda virus [36], hantavirus [37]. Shrews are known to harbor other henipaviruses including Langya virus [8]. In addition, shrews have been found to spread viruses like mammarenavirus and hepatitis E virus [38]. This raises the possibility of co-infections with both hantaviruses and henipaviruses in individual animals, creating opportunities for viral interaction, recombination, or increased transmission potential.

Additionally, prior surveillance has implicated *B. brevicauda* as a possible reservoir for Powassan virus, a member of the Orthoflavivirus genus that can cause fatal encephalitis in humans [39]. The capacity of this single species to harbor such a diverse array of potentially dangerous viruses emphasizes the importance of monitoring its role in zoonotic transmission cycles.

Our study revealed that the attachment glycoprotein, fusion protein, X protein, and matrix protein of CHV possess transmembrane domains, indicating their localization to or interaction with host cell membranes. This structural feature is characteristic of viral proteins involved in critical steps of the infection cycle, such as host cell attachment, membrane fusion, and viral assembly. Because these proteins are exposed on the viral surface or play key roles in viral entry and propagation, they are accessible to the host immune system and therefore represent promising targets for vaccine development. Their immunogenic potential, combined with their functional importance, supports further investigation into their suitability as vaccine candidates in preclinical and clinical studies.

Table 3 presents the percent identity of specific CHV proteins in comparison to corresponding proteins from other viruses. The sequence similarity analysis between CHV and other related Henipavirus species—including Sollieres shrew parahenipavirus, Ninorex virus, Melian virus, Lechodon virus, and Langya virus—provides insight into CHV's genetic and evolutionary relationship within the Henipavirus genus. The relatively low sequence identity of 35% for the attachment glycoprotein, fusion protein, and X protein suggests substantial genetic divergence in regions often associated with host specificity and immune recognition, indicating that CHV may have evolved distinct mechanisms for host entry and immune evasion. Moderate similarity in the fusion (approximately 55%) and nucleocapsid proteins (approximately 57%) points to some degree of conserved functional roles, particularly in viral replication and genome encapsidation. The matrix protein shows the highest similarity at 75%, suggesting strong evolutionary conservation in structural and assembly-related functions. The phosphoprotein's approximately 65% similarity further supports a moderate level of conservation in

Table 3. Percent identity of proteins specific to CHV proteins calculated by pairwise BLAST.

| Camp Hill virus | Sollieres shrew parahenipavirus | Ninorex virus | Melian virus | Lechodon virus | Langya virus |
|---|---|---|---|---|---|
| Attachment glycoprotein | 36.03% | 33.40% | 33.00% | 34.11% | 32.33% |
| Fusion protein | 54.91% | 55.73% | 52.04% | 52.14% | 54.07% |
| X protein | 36.90% | 32.76% | 36.36% | 34.88% | – |
| C protein | – | 38.60% | 30.12% | – | 29.63% |
| Matrix protein | 76.47% | 76.18% | 73.53% | 72.94% | 73.82% |
| Nucleocapsid protein | – | 50.73% | 57.27% | 57.17% | 59.35% |
| Phosphoprotein | – | 74.62% | 64.90% | 64.43% | 63.16% |

replication-associated components. Overall, CHV appears to be genetically distinct from other Henipaviruses, yet retains conserved elements in key structural and replication proteins, indicating that it is a novel but evolutionarily related member of the Henipavirus genus.

In this paper, we presented a comprehensive analysis of the structural composition of the proteins associated with CHV. By utilizing advanced techniques in molecular biology and protein modeling, we were able to elucidate the intricate architecture of key viral proteins, shedding light on their functional roles in the viral life cycle and host interaction. Our findings not only enhance the current understanding of CHV at a molecular level but also hold significant implications for the development of targeted diagnostic tools and effective vaccines. This foundational knowledge paves the way for future research aimed at controlling and preventing CHV infections in humans.

## Acknowledgments

The author acknowledges Abraham Thomas Foundation for providing the software resources for this work.

## Author contributions

**Conceptualization:** Sunil Thomas.

**Data curation:** Sunil Thomas.

**Formal analysis:** Sunil Thomas.

**Funding acquisition:** Sunil Thomas.

**Investigation:** Sunil Thomas.

**Methodology:** Sunil Thomas.

**Project administration:** Sunil Thomas.

**Resources:** Sunil Thomas.

**Software:** Sunil Thomas.

**Supervision:** Sunil Thomas.

**Validation:** Sunil Thomas.

**Visualization:** Sunil Thomas.

**Writing – original draft:** Sunil Thomas.

**Writing – review & editing:** Sunil Thomas.

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
