## [Decision Letter · Decision Letter 0]

18 Jul 2025

PONE-D-25-23454THE STRUCTURE OF THE PROTEINS OF CAMP HILL VIRUSPLOS ONE

Dear Dr. Thomas,

Thank you for submitting your manuscript to PLOS ONE. After careful consideration, we feel that it has merit but does not fully meet PLOS ONE’s publication criteria as it currently stands. Therefore, we invite you to submit a revised version of the manuscript that addresses the points raised during the review process. **"The risk of henipavirus outbreaks is increasing due to human activities such as deforestation, which forces greater interaction between humans and bats, the primary zoonotic reservoir hosts.": there are no references to support this statement. More references should be cited, with this one (PMID: 38042947) as an example (citing is optional).**

We look forward to receiving your revised manuscript.

Kind regards,

Benjamin M. Liu, MBBS, PhD, D(ABMM), MB(ASCP)

Academic Editor

PLOS ONE

**Journal Requirements:**

1. When submitting your revision, we need you to address these additional requirements. Please ensure that your manuscript meets PLOS ONE's style requirements, including those for file naming. The PLOS ONE style templates can be found at https://journals.plos.org/plosone/s/file?id=wjVg/PLOSOne_formatting_sample_main_body.pdf and https://journals.plos.org/plosone/s/file?id=ba62/PLOSOne_formatting_sample_title_authors_affiliations.pdf 2. Thank you for stating the following in the Acknowledgments Section of your manuscript: The author acknowledges Abraham Thomas Foundation for providing the resources for this work. We note that you have provided funding information that is not currently declared in your Funding Statement. However, funding information should not appear in the Acknowledgments section or other areas of your manuscript. We will only publish funding information present in the Funding Statement section of the online submission form. Please remove any funding-related text from the manuscript and let us know how you would like to update your Funding Statement. Currently, your Funding Statement reads as follows: The author(s) received no specific funding for this work.Please include your amended statements within your cover letter; we will change the online submission form on your behalf. 3. Thank you for stating the following in your Competing Interests section:   “None ” Please complete your Competing Interests on the online submission form to state any Competing Interests. If you have no competing interests, please state "The authors have declared that no competing interests exist.", as detailed online in our guide for authors at http://journals.plos.org/plosone/s/submit-now  This information should be included in your cover letter; we will change the online submission form on your behalf. 4. Please provide a complete Data Availability Statement in the submission form, ensuring you include all necessary access information or a reason for why you are unable to make your data freely accessible. If your research concerns only data provided within your submission, please write "All data are in the manuscript and/or supporting information files" as your Data Availability Statement. 5. When completing the data availability statement of the submission form, you indicated that you will make your data available on acceptance. We strongly recommend all authors decide on a data sharing plan before acceptance, as the process can be lengthy and hold up publication timelines. Please note that, though access restrictions are acceptable now, your entire data will need to be made freely accessible if your manuscript is accepted for publication. This policy applies to all data except where public deposition would breach compliance with the protocol approved by your research ethics board. If you are unable to adhere to our open data policy, please kindly revise your statement to explain your reasoning and we will seek the editor's input on an exemption. Please be assured that, once you have provided your new statement, the assessment of your exemption will not hold up the peer review process. 6. If the reviewer comments include a recommendation to cite specific previously published works, please review and evaluate these publications to determine whether they are relevant and should be cited. There is no requirement to cite these works unless the editor has indicated otherwise. 

Reviewers' comments:

Reviewer's Responses to Questions

**Comments to the Author**

1. Is the manuscript technically sound, and do the data support the conclusions?

Reviewer #1: Partly

Reviewer #2: Yes

2. Has the statistical analysis been performed appropriately and rigorously? 

Reviewer #1: Yes

Reviewer #2: Yes

3. Have the authors made all data underlying the findings in their manuscript fully available?

Reviewer #1: Yes

Reviewer #2: Yes

4. Is the manuscript presented in an intelligible fashion and written in standard English?

Reviewer #1: Yes

Reviewer #2: Yes

5. Review Comments to the Author

**Reviewer #1:**  Reviewer:

Comments to the Author

In the current manuscript, Thomas et al, presents a structural and immunoinformatic analysis of the proteins of Camp Hill virus (CHV), a newly identified henipavirus in North America. The subject is timely and relevant, given the zoonotic potential of henipaviruses. The integration of structural prediction tools and epitope mapping offers a valuable contribution to early vaccine design efforts. However, the manuscript requires substantial revisions in terms of scientific rigor, clarity, and novelty justification.

Major comments :-

1. While the identification of CHV is interesting, the manuscript fails to clearly define the novel scientific insights derived from this study. It is largely descriptive, with little comparative or functional interpretation

2. The structural analyses and epitope mapping are standard computational exercises. Without experimental validation, it is difficult to judge the translational potential

3. The study heavily depends on AlphaFold and I-TASSER predictions. Confidence metrics (e.g., low C-scores) suggest many models are unreliable

4. Epitope predictions (IEDB) are speculative without immunological assays or population coverage assessment.

5. No investigation is made into how the predicted epitopes might be processed or presented in the context of actual MHC molecules or host immune systems.

6. The manuscript has several grammatical errors, awkward phrasing (e.g., “CHV is not infected in humans”), and repetitive expressions (e.g., “The predicted structure provides insights…” is repeated verbatim)

7. Figures and tables are cited inconsistently and sometimes lack proper legends or interpretation.

8. The discussion repeats large portions of the introduction

9. A critical appraisal of results is missing, such as why certain proteins share higher identity with other henipaviruses, and implications for cross-reactivity or zoonotic potential

Minor comments:-

1. Table 3 should include a clearer explanation of how similarity scores were calculated (e.g., pairwise BLAST or Clustal Omega?)

2. Figures 1–8: Consider integrating the B-factor and solvent accessibility plots into a single composite figure per protein

3. The Data Availability Statement should specify if model PDB files will be made public

4. Abbreviations should be defined consistently on first use (e.g., RMSD, TM-score). Consider preparing an abbreviation table.

**Reviewer #2:**  Nipah virus has the potential for recurrent outbreaks and a high fatality rate. Currently, there are no effective preventive vaccines or specific antiviral drugs available, and treatment primarily relies on systemic symptomatic supportive therapy. Nipah virus has been included by the World Health Organization in its Priority Pathogens Blueprint List, necessitating monitoring and preparedness measures to prevent a pandemic.

The paper presents structural analysis and functional predictions of Camp Hill virus (CHV), a newly discovered Henipavirus, first reported in North America. A systematic structural analysis was conducted on the viral attachment glycoprotein, fusion protein, X protein, C protein, matrix protein, nucleocapsid protein, phosphoprotein, and RNA polymerase, focusing on the presence and location of viral proteins, domains and spatial conformation of receptor interactions and immune recognition, B-cell and T-cell epitopes, structural feature analysis, and recognition regions of antibody and immune effector.

The paper is logically clear, methodologically systematic, and conclusive. It provides structural biology reference data for understanding the possibility of human infection with CHV, comprehending its potential transmission routes and those of other shrew-borne viruses, formulating corresponding prevention strategies including vaccines, and identifying potential targets for drug development.

Although a study does not need to be fully reflected in one single paper, virus structure prediction articles, such as this paper, which analyzes from a bioinformatics perspective, do not have any major flaws. However, from a virology perspective, it is still hoped that the corresponding prediction conclusions can be verified in terms of protein function at levels of the cellular or corresponding model animals especially.

In addition, please ensure consistency in the number of decimal places retained for the percentages in Table 3.

6. PLOS authors have the option to publish the peer review history of their article (what does this mean? ). If published, this will include your full peer review and any attached files.

**Do you want your identity to be public for this peer review?** For information about this choice, including consent withdrawal, please see our Privacy Policy .

Reviewer #1: **Yes: ** Pratibha Gaur

Reviewer #2: No

---

## [Author Response · Author response to Decision Letter 1]

4 Aug 2025

Review Comments to the Author

Reviewer #1: Reviewer:

Comments to the Author

In the current manuscript, Thomas et al, presents a structural and immunoinformatic analysis of the proteins of Camp Hill virus (CHV), a newly identified henipavirus in North America. The subject is timely and relevant, given the zoonotic potential of henipaviruses. The integration of structural prediction tools and epitope mapping offers a valuable contribution to early vaccine design efforts. However, the manuscript requires substantial revisions in terms of scientific rigor, clarity, and novelty justification.

Major comments :-

1. While the identification of CHV is interesting, the manuscript fails to clearly define the novel scientific insights derived from this study. It is largely descriptive, with little comparative or functional interpretation.

Re: The objective of this manuscript is to define the structure of the proteins of Camp Hill virus. Our study represents the first detailed structural analysis of CHV proteins, using computational analysis, and provides foundational data that will enable future functional and mechanistic studies. While we recognize the value of comparative or functional insights, such analyses—especially those requiring experimental validation—are outside the scope of the present work.

2. The structural analyses and epitope mapping are standard computational exercises. Without experimental validation, it is difficult to judge the translational potential.

Re: In our study, we employed a bioinformatics-driven approach to investigate the structural and immunological features of Camp Hill virus proteins. By computationally modeling these proteins and predicting B-cell and T-cell epitopes, we aimed to lay the groundwork for future experimental research. Although our current work is purely in silico, it serves as a foundational step toward guiding experimental design and prioritizing targets for vaccine or therapeutic development. Ultimately, we recognize that translating computational predictions into practical applications requires rigorous laboratory validation. Our study should therefore be viewed as an initial phase in a broader research pipeline, intended to inform and catalyze further empirical investigation.

3. The study heavily depends on AlphaFold and I-TASSER predictions. Confidence metrics (e.g., low C-scores) suggest many models are unreliable

Re: Current bioinformatic analysis using AlphaFold and I-Tasser have limitations. It is important to interpret these predictions with caution, especially when the models are used to infer functional or immunological characteristics. Structural uncertainty can significantly impact downstream analyses such as active site identification or epitope mapping, potentially leading to misleading conclusions. Therefore, while these computational tools are valuable for hypothesis generation and preliminary insights, experimental validation remains essential to confirm the structural integrity and biological relevance of the predicted models.

4. Epitope predictions (IEDB) are speculative without immunological assays or population coverage assessment.

Re: Epitope mapping is the first step for diagnostic and vaccine study. This study provides a foundational framework by identifying candidate epitopes that warrant further investigation. Nevertheless, translating these predictions into effective diagnostic tools or vaccine components will require rigorous in vitro and in vivo validation to confirm their immunogenicity, specificity, and relevance across different demographic groups. As such, this work should be viewed as an initial step in a broader, multi-phase research process aimed at the rational design of immunological interventions.

5. No investigation is made into how the predicted epitopes might be processed or presented in the context of actual MHC molecules or host immune systems.

Re: We acknowledge that our study does not investigate the processing or presentation of the predicted epitopes in the context of MHC molecules or host immune systems. Our primary aim was to conduct a structural analysis of Camp Hill virus (CHV) proteins using computational methods, laying the groundwork for future functional and immunological studies. While epitope prediction offers preliminary insights, we agree that experimental validation and deeper exploration of antigen processing and MHC presentation pathways are essential next steps. We hope future work will build on our findings to address these important immunological aspects.

6. The manuscript has several grammatical errors, awkward phrasing (e.g., “CHV is not infected in humans”), and repetitive expressions (e.g., “The predicted structure provides insights…” is repeated verbatim)

Re: Although CHV is not known to infect humans, direct contact or exposure to infected secretions or feces could present a risk.

I have re-written or taken out the text: “The predicted structure provides insights…”

7. Figures and tables are cited inconsistently and sometimes lack proper legends or interpretation.

Re: I have edited the text based on the suggestions.

8. The discussion repeats large portions of the introduction

Re: I have edited the Discussion section.

9. A critical appraisal of results is missing, such as why certain proteins share higher identity with other henipaviruses, and implications for cross-reactivity or zoonotic potential

Re: I have included statements on these in the Discussion section.

Minor comments:-

1. Table 3 should include a clearer explanation of how similarity scores were calculated (e.g., pairwise BLAST or Clustal Omega?)

Re: Pairwise BLAST was used to calculate similarity.

2. Figures 1–8: Consider integrating the B-factor and solvent accessibility plots into a single composite figure per protein.

Re: The figures will appear as a single composite figure in the final version of the manuscript.

3. The Data Availability Statement should specify if model PDB files will be made public

Re: The model PDB files are not made public.

4. Abbreviations should be defined consistently on first use (e.g., RMSD, TM-score). Consider preparing an abbreviation table.

Re: The quality and reliability of the predicted protein model were assessed using structural validation metrics, including the confidence score (C-score), template modeling (TM-score), and root mean square deviation (RMSD). The table is prepared based on the statement. The expansion and abbreviation are stated in the table legend.

Reviewer #2: Nipah virus has the potential for recurrent outbreaks and a high fatality rate. Currently, there are no effective preventive vaccines or specific antiviral drugs available, and treatment primarily relies on systemic symptomatic supportive therapy. Nipah virus has been included by the World Health Organization in its Priority Pathogens Blueprint List, necessitating monitoring and preparedness measures to prevent a pandemic.

The paper presents structural analysis and functional predictions of Camp Hill virus (CHV), a newly discovered Henipavirus, first reported in North America. A systematic structural analysis was conducted on the viral attachment glycoprotein, fusion protein, X protein, C protein, matrix protein, nucleocapsid protein, phosphoprotein, and RNA polymerase, focusing on the presence and location of viral proteins, domains and spatial conformation of receptor interactions and immune recognition, B-cell and T-cell epitopes, structural feature analysis, and recognition regions of antibody and immune effector.

The paper is logically clear, methodologically systematic, and conclusive. It provides structural biology reference data for understanding the possibility of human infection with CHV, comprehending its potential transmission routes and those of other shrew-borne viruses, formulating corresponding prevention strategies including vaccines, and identifying potential targets for drug development.

Although a study does not need to be fully reflected in one single paper, virus structure prediction articles, such as this paper, which analyzes from a bioinformatics perspective, do not have any major flaws. However, from a virology perspective, it is still hoped that the corresponding prediction conclusions can be verified in terms of protein function at levels of the cellular or corresponding model animals especially.

Re: The primary objective of this manuscript is to characterize the structural features of the (CHV proteins through comprehensive computational analysis. To our knowledge, this study represents the first in-depth structural investigation of CHV proteins, offering a foundational dataset that can inform and guide future functional, mechanistic, and translational research. While we acknowledge the importance of incorporating comparative and functional insights, particularly those supported by experimental validation, such analyses fall beyond the scope of the current work. Our focus here is to establish a structural framework that can serve as a critical reference point for subsequent empirical studies aiming to explore the biological roles and therapeutic potential of CHV proteins.

In addition, please ensure consistency in the number of decimal places retained for the percentages in Table 3.

Re: I have edited the decimals in the table.

---

## [Decision Letter · Decision Letter 1]

22 Aug 2025

THE STRUCTURE OF THE PROTEINS OF CAMP HILL VIRUS

PONE-D-25-23454R1

Dear Dr. Thomas,

We’re pleased to inform you that your manuscript has been judged scientifically suitable for publication and will be formally accepted for publication once it meets all outstanding technical requirements.

Kind regards,

Benjamin M. Liu, MBBS, PhD, D(ABMM), MB(ASCP)

Academic Editor

PLOS ONE

Additional Editor Comments (optional):

Reviewers' comments:

Reviewer's Responses to Questions

**Comments to the Author**

1. If the authors have adequately addressed your comments raised in a previous round of review and you feel that this manuscript is now acceptable for publication, you may indicate that here to bypass the “Comments to the Author” section, enter your conflict of interest statement in the “Confidential to Editor” section, and submit your "Accept" recommendation.

Reviewer #1: All comments have been addressed

Reviewer #2: All comments have been addressed

2. Is the manuscript technically sound, and do the data support the conclusions?

Reviewer #1: Yes

Reviewer #2: Yes

3. Has the statistical analysis been performed appropriately and rigorously? 

Reviewer #1: Yes

Reviewer #2: Yes

4. Have the authors made all data underlying the findings in their manuscript fully available?

Reviewer #1: Yes

Reviewer #2: Yes

5. Is the manuscript presented in an intelligible fashion and written in standard English?

Reviewer #1: Yes

Reviewer #2: Yes

6. Review Comments to the Author

Reviewer #1: NA

They have presented and answered the questions asked by reviewer.

Reviewer: 4

Comments to the Author

In the current manuscript, Thomas et al, presents a structural and immunoinformatic analysis of the proteins of Camp Hill virus (CHV), a newly identified henipavirus in North America. The subject is timely and relevant, given the zoonotic potential of henipaviruses. The integration of structural prediction tools and epitope mapping offers a valuable contribution to early vaccine design efforts. However, the manuscript requires substantial revisions in terms of scientific rigor, clarity, and novelty justification.

Major comments :-

1. While the identification of CHV is interesting, the manuscript fails to clearly define the novel scientific insights derived from this study. It is largely descriptive, with little comparative or functional interpretation

2. The structural analyses and epitope mapping are standard computational exercises. Without experimental validation, it is difficult to judge the translational potential

3. The study heavily depends on AlphaFold and I-TASSER predictions. Confidence metrics (e.g., low C-scores) suggest many models are unreliable

4. Epitope predictions (IEDB) are speculative without immunological assays or population coverage assessment.

5. No investigation is made into how the predicted epitopes might be processed or presented in the context of actual MHC molecules or host immune systems.

6. The manuscript has several grammatical errors, awkward phrasing (e.g., “CHV is not infected in humans”), and repetitive expressions (e.g., “The predicted structure provides insights…” is repeated verbatim)

7. Figures and tables are cited inconsistently and sometimes lack proper legends or interpretation.

8. The discussion repeats large portions of the introduction

9. A critical appraisal of results is missing, such as why certain proteins share higher identity with other henipaviruses, and implications for cross-reactivity or zoonotic potential

Minor comments:-

1. Table 3 should include a clearer explanation of how similarity scores were calculated (e.g., pairwise BLAST or Clustal Omega?)

2. Figures 1–8: Consider integrating the B-factor and solvent accessibility plots into a single composite figure per protein

3. The Data Availability Statement should specify if model PDB files will be made public

4. Abbreviations should be defined consistently on first use (e.g., RMSD, TM-score). Consider preparing an abbreviation table.

Reviewer #2: Thanks the authors for their prompt response to the review comments.

The limitations of the experimental scope and the detailed presentation of accurate data reflect the rigorous logic of the paper and the authors' comprehensive understanding of the data.

The analysis of the structure of the newly discovered Henipavirus in North America presented in this paper may promote research on the pathogenicity, immunogenicity, and vaccine preparation of the virus for prevention and treatment.

7. PLOS authors have the option to publish the peer review history of their article (what does this mean? ). If published, this will include your full peer review and any attached files.

**Do you want your identity to be public for this peer review?** For information about this choice, including consent withdrawal, please see our Privacy Policy .

Reviewer #1: No

Reviewer #2: No

---

## [Editor Report · Acceptance letter]

PONE-D-25-23454R1

PLOS ONE

Dear Dr. Thomas,

I'm pleased to inform you that your manuscript has been deemed suitable for publication in PLOS ONE. Congratulations! Your manuscript is now being handed over to our production team.

Kind regards,

on behalf of

Dr. Benjamin M. Liu

Academic Editor

PLOS ONE